# Multimodal emotion recognition via adaptive high-order transforme network

Yuanyuan Lu[1,2]*, Hao Feng[2]

**1** Faculty of Artificial Intelligence in Education, Central China Normal University, Wuhan, China, **2** School of Information Engineering, Wuhan College, Wuhan, China

* yuanyuan.lu@whxy.edu.com

## Abstract

Multimodal emotion recognition leverages multiple modalities to capture emotional cues more comprehensively, thereby improving the accuracy and robustness of emotion recognition. From the perspective of multimodal data and feature learning, reducing information redundancy in multimodal data and enhancing the discriminability of deep feature co-learning can effectively boost recognition performance. Based on this, this paper proposes a multimodal emotion recognition method based on an Adaptive High-order Transformer Network (AHOT). This method constructs Adaptive Selection Transformer block (AST) and Cross-modal Feature Fusion block (CMFF) for each modality branch, aiming to fully capture non-redundant feature representations from each modality and the interactions between modalities. In addition, a sparse high-order feature learning module is designed to enable the learning of highly discriminative high-order features across modalities. Experimental results on two multimodal emotion recognition datasets (IEMOCAP and CMU-MOSEI) demonstrate that, compared with several related methods, the proposed AHOT effectively improves emotion recognition accuracy. Moreover, ablation studies and parameter analyses further validate the effectiveness of AHOT.

## Introduction

Emotion recognition [1–5] is central to social communication, as accurately identifying emotions is essential for enhancing communication efficiency. People express their emotions through various channels, including language, facial expressions, and body postures. Due to the complexity of human emotions, a single modality is often insufficient for accurately recognizing a person's emotional state. Multimodal emotion recognition [6–8] involves integrating multiple input modalities—such as audio, text, and images—to effectively model emotional information and capture the nuances of human emotional expression. Multimodal emotion recognition can effectively identify human emotional states (such as happiness, anger, sadness, surprise, etc.), providing more comprehensive and accurate emotion analysis. It enhances the quality

**Data availability statement:** The data underlying the results presented in the study are available from https://git-code.com/open-source-toolkit/d7271/commit/63d2f91352d852109657b-7d92e7e69c9ed141f1d?ref=main (IEMOCAP Dataset) and https://github.com/CMU-MultiComp-Lab/CMU-MultimodalSDK (CMU-MOSEI Dataset).

**Funding:** The author(s) received no specific funding for this work.

**Competing interests:** The authors have declared that no competing interests exist.

of human-computer interaction in complex scenarios and has already been widely applied in various fields [9–12], such as intelligent customer service, educational systems, and intelligent driving. In addressing the task of multimodal emotion recognition, several challenges arise from the perspectives of modality data and feature learning, such as:

(1) Intra-modal redundancy: Each individual modality may contain irrelevant or repetitive features, such as emotion-neutral stop words in text or background information in images.

(2) Feature discriminability: Some modalities inherently express emotion weakly (e.g., static facial images), leading to insufficiently discriminative unimodal features; additionally, the overlapping feature distributions across different emotional categories make it difficult for models to distinguish between them.

In recent years, deep learning algorithms [13–16] such as Convolutional Neural Networks (CNNs), Residual Networks (ResNets), and Long Short-Term Memory (LSTM) networks have been widely applied to research on speech signal feature extraction and speech emotion classification. Meanwhile, a large number of emotion recognition algorithms based on multimodal data have been proposed. For instance, Xu et al. proposed GateM2Former [17], which integrates MoME and MixMoE with a gating mechanism to select relevant pre-trained representations. MoME captures modality-specific characteristics, MixMoE models cross-modal interactions, and a hierarchical merge structure effectively processes long sequences such as speech. Based on transformer-based self-supervised feature fusion style, Siriwardhana et al. proposed a multimodal emotion recognition framework [18]. This framework employs pre-trained self-supervised networks to extract features from multiple modalities, including text, audio, and visual data. Additionally, it utilizes a method based on transformers and attention mechanisms to obtain contextual semantic connections and patterns. Li et al. introduced SIA-Net (Sparse Interactive Attention Network) for multimodal emotion recognition [19], using sparse intramodal and intermodal attention to enhance fusion by emphasizing key features and reducing redundancy through selective attention with minimal nonzero weights. Moreover, the proposed SDT model [20] utilized transformers and self-distillation to enhance intra/inter-modal learning and dynamic fusion. It achieved superior performance on IEMOCAP and MELD compared to previous baselines.

The aforementioned methods can effectively improve the performance of emotion recognition, but they often overlook intra-modal information redundancy and the collaborative learning of discriminative deep features. To address this, this paper proposes a multimodal emotion recognition method based on an Adaptive High-order Transformer Network. This approach accounts for feature redundancy within different modalities and adaptively captures effective feature representations. Meanwhile, a high-order feature learning module is embedded to enhance the robustness of deep features. The main contributions of this paper are reported as follows:

(1) We design an adaptive selection Transformer block and a cross-modal feature fusion block, and we embed them into a three-branch feature learning network for text, speech, and video, enabling the adaptive capture of effective feature representations within each modality.

(2) We introduce a high-order feature learning module with sparse characteristics, which not only facilitates multimodal feature fusion but also enables high-order discriminative feature learning across modalities.

(3) Experiments conducted on two challenging multimodal emotion recognition datasets demonstrate that the proposed method significantly improves recognition performance. Moreover, ablation studies validate the effectiveness of each module.

## Related work

Multimodal data often exhibit both complementary and conflicting characteristics, posing a challenge for multimodal speech emotion recognition [21–25]: how to effectively integrate the correlations and complementarities across different modalities. A typical multimodal emotion recognition framework comprises three core components: feature extraction and representation, feature fusion, and classifier optimization. Existing methods [26–31] can be broadly divided into two categories: those based on fusion strategies and those grounded in representation learning. Fusion-based approaches aim to construct sophisticated fusion mechanisms to derive joint representations from multiple modalities. In contrast, representation learning approaches focus on capturing fine-grained semantic cues within each modality, which convey rich and nuanced emotional information. Such semantic representations can further enhance the effectiveness of multimodal fusion, especially in modeling inter-modal relationships. Both approaches ultimately seek to strengthen multimodal feature representations through deep feature learning. Then Yang et al. [32] proposed a multimodal framework—Two-Phase Multi-Task Sentiment Analysis (TPMSA). It adopted a two-phase training strategy to fully leverage pre-trained models and introduced a novel multi-task learning strategy to explore the classification capabilities of each representation, while simultaneously conducting intra-modal and inter-modal interactions. Guo et al. [33] combined bidirectional LSTM and multi-head attention mechanisms, using bidirectional LSTM, convolutional neural networks, and the RoBERT model to extract features from audio, video, and text data, respectively. The features from the three modalities are then directly fed into the multi-head attention mechanism for fusion and classification. Yang et al. [34] employed adversarial learning to decouple modal features and utilized cross-modal attention for modality fusion.

In recent years, the Transformer model [35] has demonstrated unique advantages in modeling long-term sequences, leading to its widespread application in multimodal emotion recognition. Inspired by the Transformer architecture, Wang et al. [36] proposed a novel fusion method called Trans-Modality, which integrates speech, text, and visual features through Transformer-based feature learning and fusion. The final classification is performed using fully connected layers, resulting in significantly improved classification performance. Zhao et al. [37] enhanced multimodal emotion recognition using three key modules: ME (Multimodal Embedding) leverages pretrained models to mitigate limited labeled data, MT (Mutual Transformer) captures shared and speaker-specific emotional cues, and DST (Deep-Scale Transformer) refines recognition by aligning modalities and extracting multiscale features via shared GRUs. This design strengthened contextual understanding and boosts performance across diverse emotional expressions. John et al. [38] utilized the video self-attention, the audio self-attention, and the audio-video cross attention to enhance the performance of audio-visual emotion recognition. Khan et al. [39] addressed emotion recognition by introducing a cross-modal transformer that captures both detailed and broad patterns from speech and text. It employed HuBERT for audio feature extraction and BERT for textual understanding. Lian et al [40] proposed a fine-grained multimodal information fusion method based on the Transformer architecture, named conversational transformer network. Zhang et al. [41] proposed a method for simulating emotional coherence and introduced the DEAN (Deep Emotional Arousal Network) model, which includes three key components: a multimodal development module that simulates the cognitive comparator; a cross-modal Transformer that simulates the

human perceptual analysis system, and a multimodal gating block that mimics the activation mechanism of the human emotional arousal model, enabling both inter-modal and intra-modal interactions.

The aforementioned methods leverage the self-attention mechanism for multimodal feature fusion and representation learning, but they lack consideration of intra-modal information redundancy and feature discriminability. Therefore, this paper proposes an Adaptive High-order Transformer Network to acquire effective feature representations within each modality and enhance the discriminability of fused features across modalities.

## Materials and methods

### Adaptive high-order transformer network

As shown in "Fig 1", the Adaptive High-order Transformer network (AHOT) proposed in this paper mainly consists of three branches: video, audio, and text. The data from these three branches first undergo feature learning through Conv1D convolutional layers and BiLSTM. Subsequently, an Adaptive Selection Transformer block (AST) and a Cross-Modal Feature Fusion block (CMFF) are employed for non-redundant feature learning and cross-modal feature capture. Finally, sparse

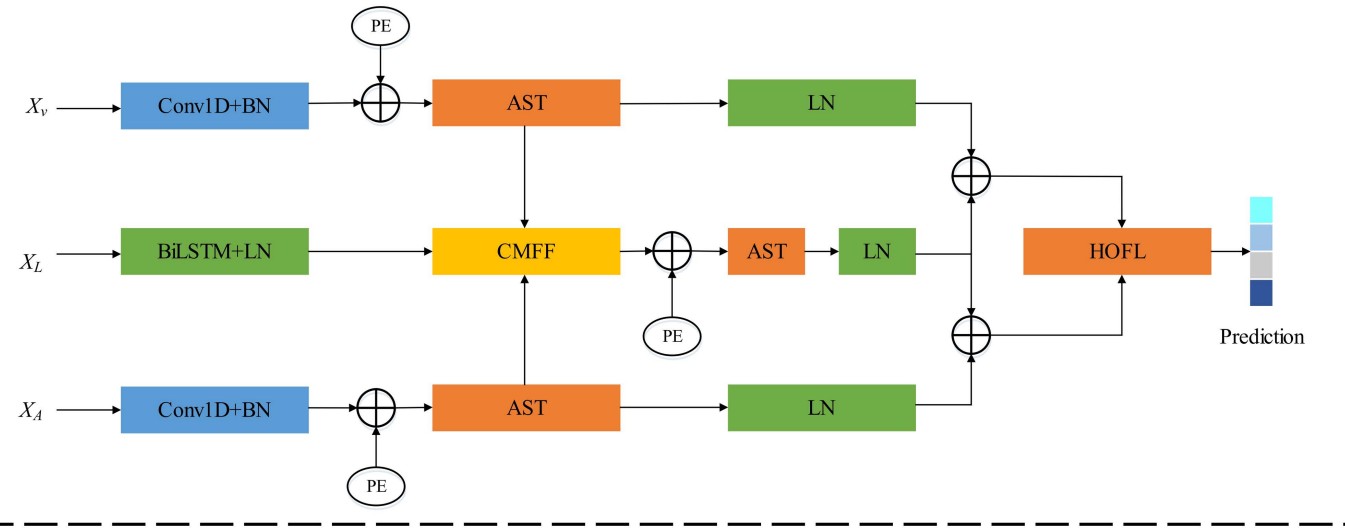

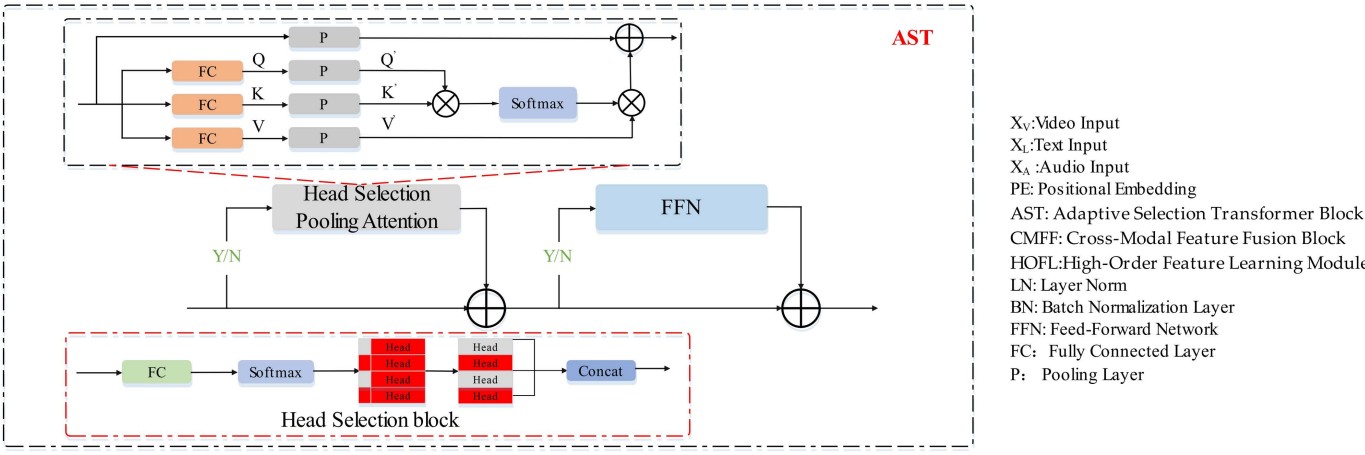

$X_V$:Video Input
$X_L$:Text Input
$X_A$ :Audio Input
PE: Positional Embedding
AST: Adaptive Selection Transformer Block
CMFF: Cross-Modal Feature Fusion Block
HOFL:High-Order Feature Learning Module
LN: Layer Norm
BN: Batch Normalization Layer
FFN: Feed-Forward Network
FC：Fully Connected Layer
P：Pooling Layer

**Fig 1. The architecture of adaptive high-order transformer network.**

high-order feature statistics are embedded into the fused feature learning mechanism to construct a High-Order Feature Learning module (HOFL), thereby acquiring discriminative feature representations.

## Adaptive selection transformer block

As the number of stacked layers in the Transformer increases, the attention areas between the attention heads tend to overlap, leading to information redundancy. The adaptive attention selection mechanism designed in this paper achieves dynamic optimization of the allocation of attention resources.

In the l-th block of Transformer, let the input feature be $\mathbf{a}_l$, and first generate the intermediate feature representation through a layer of linear transformation, as shown,

$$\mathbf{p}_l = \mathbf{W}_l \mathbf{a}_l, \quad s.t. \mathbf{p}_l \in \mathbb{R}^H, \tag{1}$$

where $\mathbf{W}_l$ is the linear transformation matrix and $H$ is the number of attention heads. In this work, the value of $H$ is 16. The linearly transformed $\mathbf{p}_l$ is used as the basis for subsequent sampling and decision-making. Then, in order to realize the differentiable sampling operation, the Gumbel-Softmax approximation is introduced in this paper to transform the decision problem into a probabilistic choice. For the first attention head, the selection probability is generated in the following way, as shown,

$$P_{l,i,k} = \frac{\exp(p_{l,i,k} + G_{l,i,k}/\tau)}{\sum\limits_{j=0}^{1} \exp(p_{l,i,k} + G_{l,i,j}/\tau)}, k \in \{0, 1\}, \tag{2}$$

where $\mathbf{G}_{l,i}$ is the Gumbel noise term, defined as $G_{l,i} = -\log(Exp_{l,i})$, where $Exp_{l,i}$ is a random variable sampled from the exponential distribution, and $\tau$ is the temperature parameter, which is used to control the smoothness of the output distribution. In this way, the "retention" or "deprecation" decision of approximately discrete attention heads can be implemented during the training stage, while maintaining gradient transitivity. After the probability calculation is completed, a fixed threshold of 0.5 is set in this paper, and the final binarization strategy matrix is determined by using the hard threshold selection method, as shown,

$$\mathbf{P}_l = \left[ \Theta\left(P_{l,1,1}\right); \Theta\left(P_{l,2,1}\right); \ldots; \Theta\left(P_{l,H,1}\right) \right], \tag{3}$$

In this paper, two attention head processing strategies are proposed, namely "partial deprecation" and "complete deprecation", to achieve more efficient feature modeling. For the "partial deprecation", the corresponding attention head is not directly deprecated, but the interference of the invalid head is weakened by replacing the generated attention matrix with the form of the identity matrix, and the coherence of the feature pathway is maintained. Specifically, when the i-th attention head of the l-layer is marked as "partially deprecated", its attention calculation process is defined as follows,

$$A(\mathbf{Q}, \mathbf{K}, \mathbf{V})_{l,i} = \begin{cases} Soft\max\left(\frac{\mathbf{Q}\mathbf{K}^T}{\sqrt{d_k}}\right) \cdot \mathbf{V}, & P_{l,i} = 1 \\ 1 \cdot \mathbf{V}, & P_{l,i} = 0 \end{cases}, \tag{4}$$

where $\mathbf{Q}$, $\mathbf{K}$, and $\mathbf{V}$ represent the query, key, and value matrices, respectively, and $d_k$ is the dimension of the key vector. In contrast, for the "complete deprecation" strategy, the attention heads marked as invalid are directly skipped in the multi-head self-attention module, and only the filtered attention heads are retained for splicing and subsequent calculation. Specifically, the long-headed attention calculation at layer $l$ -th is shown as follows,

$$MHSA(\cdot)_{l,i} = Concat\left([A_{l,i:1 \rightarrow H}ifP_{l,i}]\right) \mathbf{W}_l^{O'},$$ (5)

where $H$ represents the number of all attention heads, $\mathbf{W}_l^{O'}$ is the output projection matrix, and only the selected attention heads are spliced and mapped.

In addition to the screening of attention heads, this paper further introduces the adaptive Transformer block selection mechanism. When a Transformer block is detected to have weak representation capabilities, you can selectively skip the calculation of the block to reduce redundancy overhead. In order to improve the flexibility of choice decision, the dimension of the Transformer block selection strategy matrix $\mathbf{e}_l$ (obtained by a similar way to $\mathbf{p}_l$ in Equation 1) is extended from 1 to 2, corresponding to the Multi-Head Self-Attention (MHSA) sublayer and the FeedForward Network (FFN) sublayer inside each Transformer block, respectively. The calculation process is shown as follows,

$$\mathbf{Z}_l' = \mathbf{E}_{l,0} \cdot MHSA\left(\mathbf{Z}_l\right) + \mathbf{Z}_l,$$ (6)

$$\mathbf{Z}_{l+1} = \mathbf{E}_{l,1} \cdot FFN\left(\mathbf{Z}_l'\right) + \mathbf{Z}_l',$$ (7)

Among them, $\mathbf{E}_{l,0}$ and $\mathbf{E}_{l,1}$ control the execution of the MHSA layer and FFN, respectively. Through the adaptive Transformer block selection mechanism, the calculation path can be dynamically adjusted according to the input characteristics to improve the overall inference efficiency.

## Cross-modal feature fusion block

Assuming the three-branch output features are $\mathbf{Y}_V$, $\mathbf{Y}_L$, and $\mathbf{Y}_A$, cross-modal feature fusion block employs 1 × 1 convolution and average pooling operations to further learn semantic features from the three branches, as shown,

$$\begin{cases} \mathbf{Y}_V' = Conv\left(Avgpool\left(\mathbf{Y}_V\right)\right) \\ \mathbf{Y}_L' = Conv\left(Avgpool\left(\mathbf{Y}_L\right)\right), \\ \mathbf{Y}_A' = Conv\left(Avgpool\left(\mathbf{Y}_A\right)\right) \end{cases}$$ (8)

Subsequently, $\mathbf{Q}_c$, $\mathbf{K}_c$, and $\mathbf{V}_c$ are obtained through linear layers, and the output of the multi-scale cross-attention based on cosine similarity can be computed using the following formula:

$$\begin{cases} \tilde{Y} = \cos\left(Q_c, K_c, V_c\right) + Y_A \\ \cos\left(Q_c, K_c, V_c\right) = \frac{Q_c \cdot K_c^T}{L_Q \otimes L_K} \cdot V_c \end{cases},$$ (9)

where $\otimes$ denotes the outer product operation, $\mathbf{L}_Q$ and $\mathbf{L}_K$ represent the magnitudes (i.e., L2 norms) of $\mathbf{Q}_c$ and $\mathbf{K}_c$, $\cos\left(\cdot\right)$ denotes the computation of cosine similarity. Finally, the valuable semantic information from the three-branch features can be acquired.

## High-order feature learning module

The core idea of the High-order feature learning module is to learn a dictionary $\mathbf{A}$ with $b$ atoms from the deep feature vector feature pairs $(\mathbf{u}_i, \mathbf{v}_j)$, where each atom can be decomposed into a low-rank matrix $\mathbf{X}_i\mathbf{Y}_l^{\mathsf{T}}$. At this time, the encoding coefficient $\mathbf{c}_s$ can be calculated by the following formula:

$$\min_{\mathbf{c}_s} \left\| \mathbf{u}_i \mathbf{v}_j^\mathsf{T} - \sum_{l=1}^{k} \mathbf{r}_s^l \mathbf{X}_l \mathbf{Y}_l^\mathsf{T} \right\|^2 + \omega \|\mathbf{r}_s\|_1, \tag{10}$$

where $\omega$ is an adjustable parameter, $s = 1, 2, ..., C$, $\|\cdot\|_1$ is the $l_1$ norm operator, and $\mathbf{r}_s^l$ represents the l-th element of the encoding coefficient. $\mathbf{X}_l \in \mathsf{R}^{p \times v}$, $\mathbf{Y}_l^\mathsf{T} \in \mathsf{R}^{v \times q}$, and $v << p$. It can be seen from formula (11) that the FBC module can be solved by the LASSO (Least Absolute Shrinkage and Selection Operator) algorithm, that is:

$$\mathbf{r}_s = \operatorname{sign}(\mathbf{r}_s') \odot \max\left(\operatorname{abs}(\mathbf{r}_s') - {}^\omega\!/_2, 0\right), \tag{11}$$

where $\mathbf{r}_s' = \mathbf{Q}\left(\mathbf{X}_r^\mathsf{T} \mathbf{u}_i \odot \mathbf{Y}_r^\mathsf{T} \mathbf{v}_j\right)$, $\odot$ is the Hadamard product, and $\mathbf{Q} \in \mathsf{R}^{b \times vb}$ is a fixed binary matrix. $\mathbf{X}_l$ and $\mathbf{Y}_l$ are calculated through the low-rank matrices $\mathbf{X}$ and $\mathbf{Y}$ to reduce the computational complexity. Here, $\mathbf{X}_l$ and $\mathbf{Y}_l$ can be expressed as:

$$\begin{cases} \mathbf{X}_l = I\left(\left(p_l \mathbf{1}_{vb}^\mathsf{T}\right) \odot \mathbf{X}^\mathsf{T}\right)/v \\ \mathbf{Y}_l = \left(I\mathbf{Y}^\mathsf{T}\right)/v \end{cases}, \tag{12}$$

where $\mathbf{1}_{vb}^\mathsf{T}$ and $I$ are all-one vectors and matrices, respectively, $p_l$ is the l-th column of $\mathbf{J}$, then $\mathbf{J}$ is defined as:

$$\mathbf{J} = \left(\left(\mathbf{Q}\left(\mathbf{X}^\mathsf{T}\mathbf{X}\mathbf{Q}^\mathsf{T} \odot \mathbf{Y}^\mathsf{T}\mathbf{Y}\mathbf{Q}^\mathsf{T}\right)\right)^{-1}\mathbf{Q}\right)^\mathsf{T}, \tag{13}$$

Through the above solution, the sparse second-order attention feature vector obtained by the HOFL module can be expressed as:

$$\mathbf{F} = \max\left\{\mathbf{r}_s\right\}_{s=1}^{b}, \tag{14}$$

where $\mathbf{F} \in \mathsf{R}^{1 \times 1 \times b}$ is obtained by maximizing and traversing the aggregation of each atom in the dictionary $\mathbf{A}$, and $b << C^2$. The High-order feature learning module takes into account the feature redundancy of the second-order feature statistics and can achieve a better feature enhancement effect. Finally, a pixel-wise normalization layer is introduced to enhance the discriminability of features, which mainly includes a sign square root normalization layer and an $l_2$ normalization layer. The specific calculation formula is as follows:

$$\begin{cases} \mathbf{H}_i^{j''} = \operatorname{sign}\left(\mathbf{H}_i^{j'}\right)\sqrt{\left|\mathbf{H}_i^{j'}\right| + \varepsilon} \\ \mathbf{H}_i^{j''} = \sqrt{\sum_{j=1}^{N}\left(\mathbf{H}_i^{j'}\right)^2 + \varepsilon} \end{cases}, \tag{15}$$

where $\mathbf{H}_i^{j'}$ is the feature descriptor of the i-th row and j-th column of the feature vector $\mathbf{F}$, and $\operatorname{sign}(\cdot)$ is the sign function, that is: when $\mathbf{H}_i^{j'} > 0$, $\operatorname{sign}(\mathbf{H}_i^{j'}) = 1$; when $\mathbf{H}_i^{j'} = 0$, $\operatorname{sign}(\mathbf{H}_i^{j'}) = 0$; when $\mathbf{H}_i^{j'} < 0$, $\operatorname{sign}(\mathbf{H}_i^{j'}) = -1$. $\varepsilon$ is a small integer to ensure the meaningfulness of the formula.

## Results and discussion

### Datasets and experimental setting

To validate the effectiveness of the network proposed in this paper, we employ two challenging multimodal emotion recognition datasets: IEMOCAP [42] and CMU-MOSEI [43]. The IEMOCAP dataset, developed by the SAIL Laboratory at the

University of Southern California, contains information from audio and textual transcripts as well as facial expressions. It comprises recorded dialogues between two individuals, which have been manually annotated and verified. The dataset includes 7,433 samples, and in this study, we use the four-class emotion version (anger, sadness, happiness, and neutrality) for evaluation.

The CMU-MOSEI dataset consists of 23,453 annotated video segments from 1,000 distinct speakers, covering 250 topics. Each video segment includes manually transcribed text aligned with the audio at the phoneme level. All videos are sourced from online video-sharing platforms. Each segment is manually annotated with an emotion score ranging from -3 to 3, representing a spectrum from highly negative to highly positive.

The experiments in this study were conducted using the PyTorch framework on an Ubuntu operating system equipped with an NVIDIA GeForce GTX 4090 GPU and 64 GB of RAM. The model was optimized using the Adam optimizer, with a batch size of 32, a learning rate of $3 \times 10^{-3}$, and a dropout rate of $3.5 \times 10^{-3}$. For the textual data, the pre-trained GloVe model was employed to perform word embedding, resulting in 300-dimensional vectors. For the audio data, features were extracted using the COVAREP acoustic analysis framework, yielding a feature dimension of 74. For the visual data, facial muscle movements were captured using the FACET facial expression analysis toolkit, which represents 35 facial action units. Temperature $\tau$ in AST is set to 5, the number of atoms $b$ and the low-rank matrix dimension $v$ in HOFL are set to 5 and 4096 respectively in our paper.

## Experimental comparison

This paper conducts experiments under aligned data settings and compares the results with several related models, such as LF-LSTM [44], Graph-MFN [45], RAVEN [46], MCTN [47], MulT [48], LMF-MulT [49], and LMR-CBT [50]. The experimental settings for the above comparison algorithms follow the parameter configurations mentioned in their respective papers. Accuracy (Acc) and F1 score are adopted as quantitative evaluation metrics, where Acc7 and Acc2 represent the 7-class classification accuracy and 2-class classification accuracy, respectively. Tables 1 and 2 present the experimental results on the IEMOCAP and CMU-MOSEI datasets, respectively.

As shown in the table above, for the IEMOCAP dataset, the AHOT method achieved the best recognition performance in the "Happy", "Sad", and "Neutral" emotion categories. Specifically, for the "Happy" category, AHOT reached an accuracy of 88.3%, outperforming the LMR-CBT method by 0.4%. In the "Sad" and "Neutral" categories, AHOT achieved the highest performance in both accuracy and F1 score. Although the proposed method performed slightly worse than the best-performing RAVEN method in the "Angry" category, the accuracy and F1 score were at a comparable level. Overall, across all emotion categories, the AHOT method outperformed the other seven approaches.

**Table 1. Comparison on the IEMOCAP dataset.**

| Method | Happy | | Sad | | Angry | | Neutral | |
|---|---|---|---|---|---|---|---|---|
| | Acc (%) | F1 (%) | Acc (%) | F1 (%) | Acc (%) | F1 (%) | Acc (%) | F1 (%) |
| LF-LSTM | 85.1 | 86.3 | 78.9 | 81.7 | 84.7 | 83.0 | 67.1 | 67.6 |
| Graph-MFN | – | – | – | – | – | – | – | – |
| RAVEN | 87.3 | **85.8** | 83.4 | 83.1 | **87.3** | **86.7** | 69.7 | 69.3 |
| MCTN | 84.9 | 83.1 | 80.5 | 79.6 | 79.7 | 80.4 | 62.3 | 57.0 |
| MulT | 86.6 | 84.5 | 84.0 | 84.1 | 84.9 | 85.5 | 69.3 | 68.6 |
| LMF-MulT | 85.3 | 84.1 | 84.1 | 83.4 | 85.7 | 86.2 | 71.2 | 70.8 |
| LMR-CBT | 87.9 | 84.6 | 85.3 | 84.4 | 86.2 | 86.3 | 71.5 | 70.6 |
| AHOT (Ours) | **88.3** | 85.4 | **85.9** | **85.0** | 87.0 | 86.6 | **72.3** | **71.5** |

* The best results are highlighted in bold, -denotes that the experiment is not conducted.

**Table 2. Comparison on the CMU-MOSEI dataset.**

| Method | Acc$_7$(%) | Acc$_2$(%) | F1(%) |
|---|---|---|---|
| LF-LSTM | 48.8 | 80.6 | 80.6 |
| Graph-MFN | 45.0 | 76.9 | 77.0 |
| RAVEN | 50.0 | 79.1 | 79.5 |
| MCTN | 49.6 | 79.8 | 80.6 |
| MulT | 51.8 | 82.5 | 82.3 |
| LMF-MulT | 50.2 | 80.3 | 80.3 |
| LMR-CBT | 50.7 | 80.5 | 80.9 |
| **AHOT (Ours)** | **52.0** | **82.7** | **82.9** |

\* The best results are highlighted in bold.

For the CMU-MOSEI dataset, AHOT achieved the best results across all three quantitative evaluation metrics: Acc7, Acc2, and F1. Compared with the cross-modal Transformer-based LMR-CBT method, AHOT showed improvements of 1.3%–2.0% in various metrics. When compared with the MulT method, which uses stacked transformers for soft alignment, AHOT also demonstrated a certain degree of improvement. This can be mainly attributed to the sparse feature learning mechanism and high-order feature fusion employed in AHOT. In summary, compared to other related algorithms, the proposed AHOT method exhibits a clear advantage in quantitative evaluation metrics.

## Ablation study

To verify the effectiveness of different modules in the proposed method, ablation experiments were designed as shown in Fig 2. The results of individual module evaluations demonstrate that, on both the IEMOCAP and CMU-MOSEI datasets, CMFF effectively integrates multimodal data and achieves greater performance gains compared to AST and HOFL. Moreover, combining any two modules yields better emotion recognition performance than using a single module alone. Notably, the combination of CMFF and HOFL results in substantial improvements, further validating the effectiveness of the high-order feature fusion mechanism in multimodal emotion recognition. The best performance is achieved while all three modules are integrated. These ablation studies confirm the effectiveness of the proposed network modules and analyze the performance gains contributed by each.

## Parameter analysis

Two key parameters in the proposed HOFL—namely, the number of atoms in the dictionary and the dimensionality of the low-rank matrix—significantly affect the emotion recognition performance of the proposed network. Therefore, Figs 3 and 4 respectively present the emotion recognition results of the proposed AHOT on the IEMOCAP dataset and the CMU-MOSEI dataset under varying numbers of atoms and dimensionality parameters, using the F1 score as the quantitative evaluation metric.

As shown in the figure above, different low-rank matrix parameters and atom quantities have varying impacts on the network. In most cases, as the number of atoms increases, the network's classification accuracy first improves and then declines. Additionally, a larger low-rank matrix parameter does not necessarily lead to better network performance. These trends are largely attributed to the complexity of the network architecture and the amount of training data available. Based on the experimental results, we observed that while the number of atoms and the low-rank matrix dimension are set to 5 and 4096 respectively, AHOT achieves optimal recognition performance on both the IEMOCAP and CMU-MOSEI datasets. Therefore, this parameter configuration is adopted for all final experiments in this study.

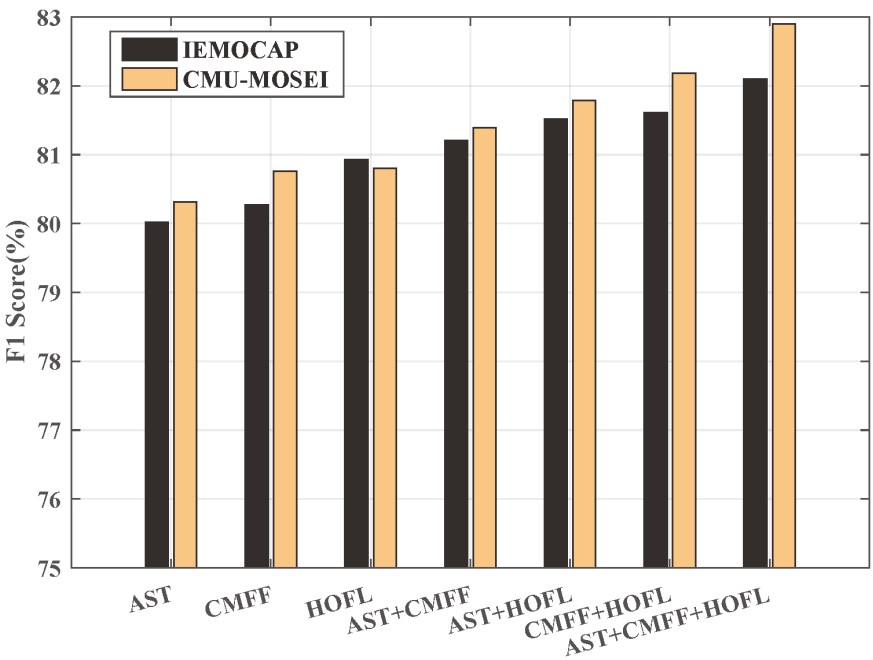

**Fig 2. Ablation study.**

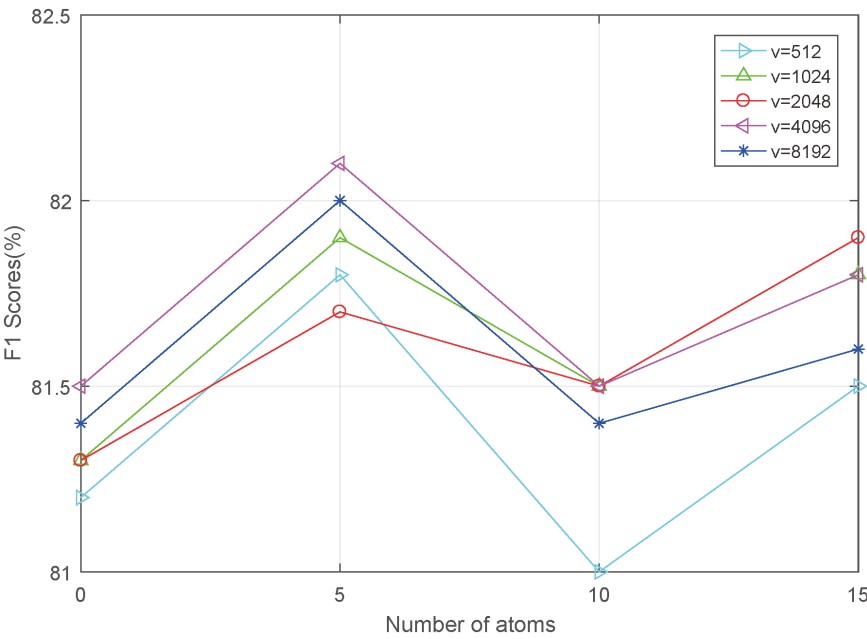

**Fig 3. The influence of number of atoms and dimensions on the IEMOCAP dataset.**

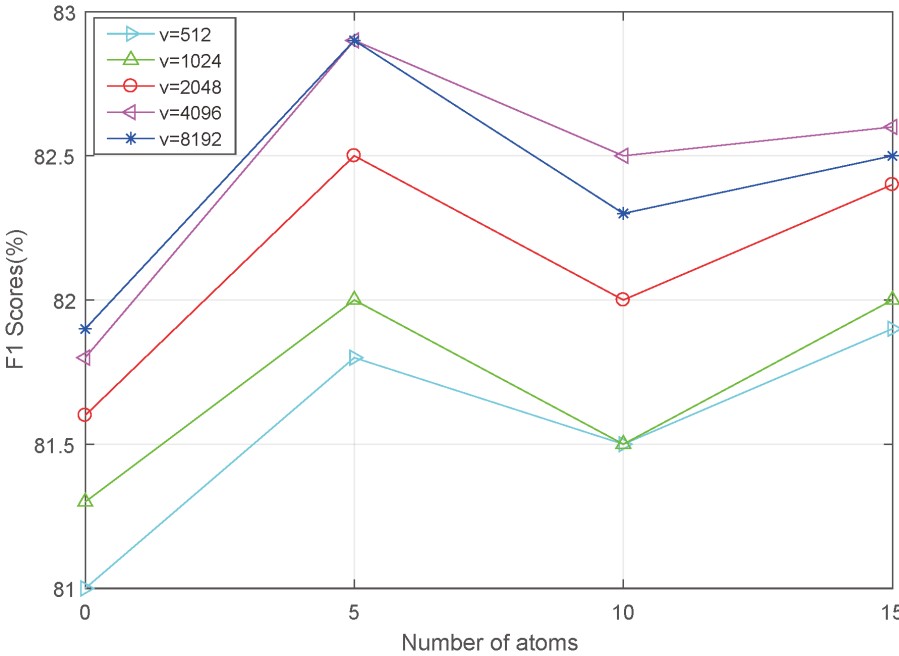

**Fig 4. The influence of number of atoms and dimensions on the CMU-MOSEI dataset.**

## Discussion

Table 3 presents the analysis of model efficiency. We adopt the T2T-ViT model as the backbone, which utilizes full self-attention heads and transformer blocks. The AST (Adaptive Sparse Transformer) mechanism proposed in this work effectively controls the number of attention heads. In our emotion recognition experiments, the number of heads was constrained to approximately 4 and 5. Experimental results show that, compared to the T2T-ViT model with 7 heads, AST achieves good recognition performance even with fewer parameters. Additionally, while the performance of models with 4 and 5 heads is nearly identical—with Head=5 yielding only a marginal improvement of 0.1%—the model with 4 heads is selected as the optimal configuration due to its lower parameter count and computational cost. In conclusion, the results demonstrate the effectiveness of the AST mechanism in reducing model redundancy. Moreover, AST demonstrates a significant advantage in terms of the accuracy-efficiency trade-off.

Fig 5 presents the emotion recognition confusion matrix on the IEMOCAP dataset test set, with the horizontal axis representing the predicted labels and the vertical axis indicating the ground truth labels. As shown in the figure, AHOT achieves recognition accuracies of 88%, 86%, 87%, and 87% for the "Happy," "Sad," "Angry," and "Neutral" emotion categories, respectively. Misclassifications of varying degrees are observed across all emotion categories. For instance, 2% of "Happy" samples are misclassified as "Sad," while 6% of "Sad" samples are misclassified as "Angry." Among all categories, "Neutral" exhibits the lowest recognition accuracy, with misclassification rates of 7%, 10%, and 11%, mainly due

**Table 3. The analysis of model efficiency.**

| Method | Head | FLOPs(G) | Parameter (M) | Total Inference Time(s) | Acc$_2$(%) |
|---|---|---|---|---|---|
| T2T-ViT[51] | 7 | 8.5 | 12 | 48.24 | **83.0** |
| AST | **4** | **3.9** | **5.5** | **21.92** | 82.7 |
| AST | 5 | 4.3 | 6.8 | 24.89 | 82.8 |

to the subtle nature of emotional cues in this category, which are difficult to accurately capture. Future work may focus on improving the AHOT network to enhance recognition performance for the "Neutral" category.

In this section, Principal Component Analysis (PCA) is employed to reduce the dimensionality of the 3D vectors to 2D, followed by t-SNE for visualizing the feature clustering on the validation set, as illustrated in Fig 6. As shown in the figure, the AHOT features not only achieve successful separation among different emotion categories but also exhibit more compact clustering. Although a few outliers are present, these cases are rare and are likely attributable to the "Neutral" category, whose subtle emotional variations are more challenging to capture.

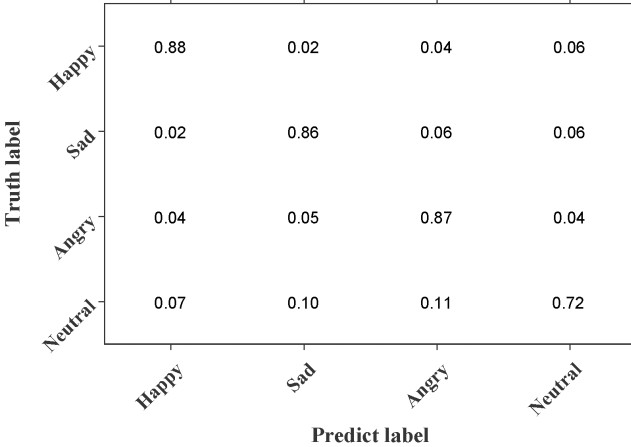

**Fig 5. The confusion matrix of the IEMOCAP dataset.**

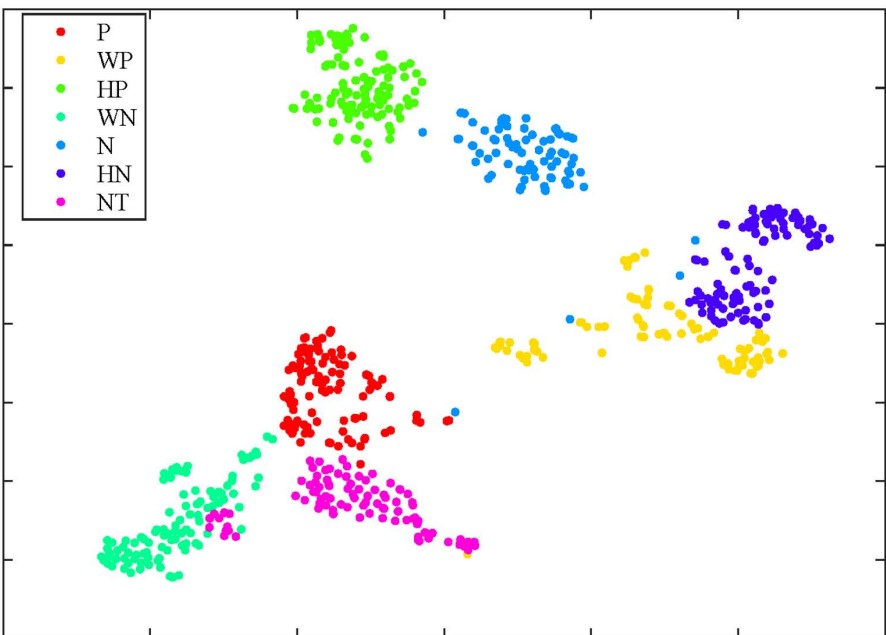

**Fig 6. t-SNE visualization on the CMU-MOSEI dataset.** P, WP, HP, WN, N, HN, and NT represent Positive, Weakly Positive, Highly Positive, Weakly Negative, Negative, Highly Negative, and Neutral, respectively.

## Conclusions

This paper proposes a novel Adaptive High-order Transformer Network (AHOT), which consists of three core components: (1) an adaptive selection transformer module for optimizing unimodal feature representations; (2) a cross-modal feature fusion module to facilitate interaction between modalities; and (3) a sparse high-order feature learning module to uncover deep discriminative features within multimodal data. Experiments conducted on the IEMOCAP and CMU-MOSEI datasets demonstrate that AHOT outperforms existing methods in emotion recognition tasks. Further ablation studies and parameter analyses confirm the contribution of each module.

Moreover, current methods may fail to adapt to varying modality importance across different scenarios. Future work could explore learnable modality weights or context-aware fusion strategies to enable dynamic adjustment of modality contributions. Additionally, leveraging large-scale unlabeled data (e.g., video–text or speech–text pairs) through contrastive learning or masked modeling approaches (such as Multimodal BERT) to learn generalizable emotional representations is another promising direction to address the challenge of limited labeled emotion recognition data.

## Author contributions

**Conceptualization:** Yuanyuan Lu.

**Data curation:** Yuanyuan Lu.

**Formal analysis:** Yuanyuan Lu.

**Methodology:** Yuanyuan Lu, Hao Feng.

**Project administration:** Hao Feng.

**Validation:** Yuanyuan Lu.

**Visualization:** Yuanyuan Lu.

**Writing – original draft:** Yuanyuan Lu, Hao Feng.

**Writing – review & editing:** Yuanyuan Lu.

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
