## [Decision Letter · Decision Letter 0]

27 Aug 2025

PONE-D-25-37321Multimodal Emotion Recognition via Adaptive High-order Transformer NetworkPLOS ONE

Dear Dr. Lu,

Thank you for submitting your manuscript to PLOS ONE. After careful consideration, we feel that it has merit but does not fully meet PLOS ONE’s publication criteria as it currently stands. Therefore, we invite you to submit a revised version of the manuscript that addresses the points raised during the review process.

We look forward to receiving your revised manuscript.

Kind regards,

Shuai Liu

Academic Editor

PLOS ONE

Journal Requirements: 

[This research was supported by the Hubei Provincial Education Science Planning Key Project (grant number 2024GA085) and 2023 Wuhan College Research Fund Program Class A Project (grant number JJA202303).]

 [The author(s) received no specific funding for this work.]

Reviewers' comments:

Reviewer's Responses to Questions

**Comments to the Author**

1. Is the manuscript technically sound, and do the data support the conclusions?

Reviewer #1: Yes

Reviewer #2: Yes

2. Has the statistical analysis been performed appropriately and rigorously? 

Reviewer #1: Yes

Reviewer #2: Yes

3. Have the authors made all data underlying the findings in their manuscript fully available?

Reviewer #1: Yes

Reviewer #2: Yes

4. Is the manuscript presented in an intelligible fashion and written in standard English?

Reviewer #1: Yes

Reviewer #2: Yes

5. Review Comments to the Author

Reviewer #1: In Figure 1, Xv,XL, and XA respectively represent the inputs of video, text, and audio? It would be best if there had a brief description to facilitate reader’ reading. What is the abbreviation of BN? Is BN the BiLSTM Network?

What are the benefits of using Conv1D in the Adaptive High-order Transformer Network designed in this paper? Especially when compared with the multi-scale convolution network commonly used in current image feature extraction.

In the experimental comparison section, it is necessary to briefly explain why only the accuracy rate and the F1 score were selected for the comparison and analysis, because there are other index parameters for the evaluation of deep learning algorithms. For examples�the precision rate, the specificity and the ROC curve.

Similarly, in the ablation experiment and parameter analysis sections, it is necessary to explain why only the F1 score was selected for the comparison and analysis.

Reviewer #2: The paper introduces a combination of adaptive attention head/layer selection and high-order sparse feature learning for multimodal emotion recognition. The proposed method is evaluated on IEMOCAP and CMU-MOSEI datasets, achieving competitive performance. The work is clearly structured and includes comparisons and ablation studies. However, there are still issues related to terminology consistency, result presentation, reproducibility details, and efficiency evidence. These are mostly minor and can be resolved with targeted revisions.

1.Terminology and notation consistency

The term “Cross-Modal Feature Fusion” is abbreviated inconsistently: the abstract uses “MCFF,” while the method section and figure captions use “CMFF.” Please unify the abbreviation throughout the paper and provide a table of abbreviations when they first appear. In the ablation study, “ATS” is used, but it should be “AST” (Adaptive Selection Transformer). Please unify this throughout the text and figure captions. In Figure 1, “Corss-Modal” should be corrected to “Cross-Modal.” Please check for consistent spelling across the paper. Equation (6) uses “MHS” but this should be “MHSA” (multi-head self-attention). Also, the temperature τ used in the selection mechanism is introduced but its range and default setting are missing. Suggestion: Add a short subsection “Notation & Abbreviations” at the beginning of the Method section with a clear abbreviation table. Correct figure captions and unify terminology. Provide a symbol table and default hyperparameters in the Appendix.

2.Inconsistencies in result reporting

On IEMOCAP, Table 1 shows category-wise accuracy (e.g., Neutral accuracy 72%, F1 71.5%), but the text discussing Figure 5 states the four-class accuracies are 88/86/87/87%. Please clarify whether these come from different splits, thresholds, or metrics, and unify the reporting. In Figure 6, the caption refers to “CMU-MOSI,” while elsewhere the dataset is called “CMU-MOSEI.” Please unify the dataset name. In Table 1, the column for Graph-MFN contains “–” for all entries. Please either provide the results for this baseline or explain in a footnote why they are unavailable/unreported.

3.Efficiency evidence is insufficient

The paper claims that adaptive head/layer selection reduces redundancy and improves efficiency, but no evidence is provided in terms of FLOPs, latency, parameter counts, or energy. At least on CMU-MOSEI, please report training/inference time, parameter counts, FLOPs, and accuracy-efficiency trade-offs compared with the non-adaptive version.

4.Expression and formatting issues

Method names: the abstract uses “adaptive head selection Transformer blocks (AST) and cross-modal feature fusion blocks (MCFF),” while the Method section titles use slightly different forms. Please unify capitalization and plurality.

References: items #15 and #41 are duplicates (“Deep emotional arousal network,” Information Fusion 2022). Please remove duplicates and recheck numbering.

6. PLOS authors have the option to publish the peer review history of their article (what does this mean? ). If published, this will include your full peer review and any attached files.

**Do you want your identity to be public for this peer review?** For information about this choice, including consent withdrawal, please see our Privacy Policy .

Reviewer #1: No

Reviewer #2: No

---

## [Author Response · Author response to Decision Letter 1]

7 Sep 2025

General comments to the reviewers and editor

First of all, thank you very much for review our paper. We carefully considered your comments. Here, we explain how we revised the paper based on those comments and suggestions. We want to extend our appreciation for taking the time and effort necessary to provide such insightful guidance.

The revision, based on the review team’s collective input, includes a number of positive changes, all changes made to the text are in red color in the paper. Based on your helpful guidance, we mainly make the following changes:

1.Analysis the effectiveness of proposed model .

2.Analysis and clarify the overall architecture and the detail of proposed block clearly.

3.Check all the figures and equations.

4.Grammar and vocabulary mistakes are corrected.

5.Spelling errors and typos are scanned and corrected thoroughly.

Finally, please refer to the attachment "Response to Reviewers" for detailed response information on all review comments.

---

## [Decision Letter · Decision Letter 1]

17 Sep 2025

Multimodal Emotion Recognition via Adaptive High-order Transformer Network

PONE-D-25-37321R1

Dear Dr. Lu,

We’re pleased to inform you that your manuscript has been judged scientifically suitable for publication and will be formally accepted for publication once it meets all outstanding technical requirements.

Kind regards,

Shuai Liu

Academic Editor

PLOS ONE

Additional Editor Comments (optional):

Reviewer #1:

Reviewer #2:

Reviewers' comments:

Reviewer's Responses to Questions

**Comments to the Author**

1. If the authors have adequately addressed your comments raised in a previous round of review and you feel that this manuscript is now acceptable for publication, you may indicate that here to bypass the “Comments to the Author” section, enter your conflict of interest statement in the “Confidential to Editor” section, and submit your "Accept" recommendation.

Reviewer #1: (No Response)

Reviewer #2: All comments have been addressed

2. Is the manuscript technically sound, and do the data support the conclusions?

Reviewer #1: (No Response)

Reviewer #2: Yes

3. Has the statistical analysis been performed appropriately and rigorously? 

Reviewer #1: (No Response)

Reviewer #2: Yes

4. Have the authors made all data underlying the findings in their manuscript fully available?

Reviewer #1: (No Response)

Reviewer #2: (No Response)

5. Is the manuscript presented in an intelligible fashion and written in standard English?

Reviewer #1: (No Response)

Reviewer #2: Yes

6. Review Comments to the Author

Reviewer #1: (No Response)

Reviewer #2: The authors have substantially revised the manuscript in response to the reviewers’ comments. They clarified the architecture, corrected terminology and spelling errors, added explanations for symbols, introduced abbreviation and hyperparameter tables, and included additional efficiency experiments (FLOPs, parameter counts, inference time). The overall readability, consistency, and completeness of the paper have improved significantly.

7. PLOS authors have the option to publish the peer review history of their article (what does this mean? ). If published, this will include your full peer review and any attached files.

**Do you want your identity to be public for this peer review?** For information about this choice, including consent withdrawal, please see our Privacy Policy .

Reviewer #1: No

Reviewer #2: No

---

## [Editor Report · Acceptance letter]

PONE-D-25-37321R1

PLOS ONE

Dear Dr. Lu,

I'm pleased to inform you that your manuscript has been deemed suitable for publication in PLOS ONE. Congratulations! Your manuscript is now being handed over to our production team.

Kind regards,

on behalf of

Dr. Shuai Liu

Academic Editor

PLOS ONE